# Gastrointestinal Digestion and Microbial Hydrolysis of Alkyl Gallates: Potential Sustained Release of Gallic Acid

**DOI:** 10.3390/foods11233936

**Published:** 2022-12-06

**Authors:** Xinmiao Wang, Qian Wang, Yuanyuan Hu, Fawen Yin, Xiaoyang Liu, Dayong Zhou

**Affiliations:** 1School of Food Science and Technology, Dalian Polytechnic University, Dalian 116034, China; 2Institute for Advanced Study, Shenzhen University, Shenzhen 518060, China; 3Development of Food Industry, Shenzhen University, Shenzhen 518060, China; 4National Engineering Research Center of Seafood, Dalian Polytechnic University, Dalian 116034, China

**Keywords:** alkyl gallates, distribution, hydrolysis, gastrointestinal tract, oral administration

## Abstract

Phenolipids such as alkyl gallates (A-GAs) have been approved by the food industry as non-toxic antioxidant additives, which are also regarded as an emerging source of functional food ingredients. However, comprehensive understanding of their digestive absorption is needed. Thus, the models of live mice and anaerobic fermentation were used to clarify the distribution and microbial hydrolysis characteristics of A-GAs in the gastrointestinal tract. HPLC-UV results demonstrated that A-GAs could be hydrolyzed by intestinal lipases and gut microorganisms including *Lactobacillus* to produce free gallic acid (GA). Through regulating the chain length of the lipid part in A-GAs, the sustained and controllable release of the GA can be easily achieved. Furthermore, A-GAs were also able to reach the colon and the cecum, which would lead to potential gastrointestinal protective effects. Therefore, A-GAs may be applied as possible ingredient for functional foods.

## 1. Introduction

A phenolipid is produced by the dehydration condensation reaction of phenolic acid (or phenol) and fatty alcohol (or fatty acid) under the catalysis of a chemical catalyst or catalyticase [1]. As a typical representative of phenolipids, alkyl gallates (A-GAs) such as propyl gallate (G-C3:0) have been approved to be used as food antioxidants in oils, fried pastas, fermented meat products, dried aquatic products and chewing gum candies [2,3]. In the European Commission, dodecyl gallate (G-C12:0) and octyl gallate (G-C8:0) are also important antioxidant in foods approved as safe additives. Moreover, A-GAs also have a variety of positive effects on health, which makes it possible to be used as a new functional ingredient. Currently, their biological properties including anticancer [4], antifungal [5], antioxidant [6] and antiviral [7] activities have been widely reported.

Due to lack of information about the in vivo digestion and absorption mechanisms, the practical application of A-GAs as ingredients of functional food is limited. To date, most studies have been focused on the quantitative detection of short-chain A-GAs and their metabolites in plasma, or their transformation in the perfused rat liver. Generally, the metabolism of short-chain A-GAs in rats and rabbits involves hydrolysis to gallic acid and further metabolism [8]. For example, Eler et al. [9], found that gallic acid (GA) was released into the systemic circulation after single intravenous injection of methyl gallate (G-C1:0) and G-C3:0. Similarly, Gao et al. [10] found that after an intragastric administration of ethyl gallate (G-C2:0) solution (20 mg/kg), both of free GA and G-C2:0 could be detected with high concentrations in rat plasma. It is worth noting that gastrointestinal distribution and microbial hydrolysis characteristics are of critical importance to the digestion, absorption and metabolism of nutrients [11]. Therefore, the related studies are necessary for explanations of the distribution and microbial hydrolysis properties of A-GAs in the gastrointestinal tract.

Our previous studies have shown that compared with orally administered tyrosol (TYr), tyrosol acyl esters (TYr-Es) can reach the large intestine (colon and cecum). In the colon and cecum, TYr-Es can be hydrolyzed by gut microorganisms to produce free TYr [12,13]. However, the ester bond in A-GAs is different from that in TYr-Es (Figure 1). In brief, TYr-Es are the condensation products of fatty acids and phenolic alcohol, while A-GAs are the condensation products of fatty alcohols and phenolic acid. Obviously, the chain lengths of their lipid moieties and structures of ester bonds will affect the digestion and absorption of phenolipids [14,15]. Therefore, the distribution and microbial hydrolysis characteristics of orally administered A-GAs in the gastrointestinal tract should be clarified.

Given this, equal molar concentrations of A-GAs including butyl gallate (G-C4:0) and myristyl gallate (G-C14:0) were administered by gavage to mice. At different time points after administration, the contents in the segments of the gastrointestinal tract including the colon, the caecum, the small intestine and the stomach were collected. The fermentation solutions consist of A-GAs with different alkyl chain lengths, and fecal slurry (or suspensions of *Lactobacillus* isolated from the mice) at different times were also obtained. Finally, the HPLC-UV measurement was used to analyze the molar concentrations of GA or A-Gas in the above gastrointestinal contents and the fermentation solutions. The related data will strongly support the possibility of A-GA’s application in functional food with various biological activities.

## 2. Materials and Methods

### 2.1. Chemicals and Reagents

GA was bought from BBI Life Sciences Co., Ltd. (Shanghai, China). Stearyl gallate (G-C18:0), dodecyl gallate (G-C12:0), propyl gallate (G-C3:0), ethyl gallate (G-C2:0) and methyl gallate (G-C1:0) were bought from Aladdin-reagent Co., Ltd. (Shanghai, China). Hexadecyl gallate (G-C16:0), octyl gallate (G-C8:0) and butyl gallate (G-C4:0) were bought from TCI Development Co., Ltd. (Shanghai, China). Myristyl gallate (G-C14:0) was bought from Toronto Research Chemicals Co., Ltd. (Toronto, ON, Canada). The purities of GA and A-GAs are all above 98%. Ultrapure water was bought from Wahaha Co., Ltd. (Hangzhou, China). Acetic acid and methanol were of HPLC grade, which were bought from Aladdin-reagent Co., Ltd. (Shanghai, China) and Spectrum Chemical Co., Ltd. (Gardena, CA, USA), respectively. All of other reagents were of analytical grade.

### 2.2. Animals

Male mice were bought from Liaoning ChangSheng Biotechnology Co. (Benxi, China), which was of Kun Ming (KM) strain weighed between 15 to 18 g. The feeding conditions were as follows: standard laboratory chow, ad libitum drinking, 12 h dark and 12 h light cycle, humidity (55 ± 5%), temperature (23 ± 2 °C) and eight mice per cage. The mice were fasted for 15–18 h before one-time oral gavage of GA or A-GAs. At different time points, the animals were sacrificed by CO_2_ asphyxiation in a covered container that attached to a CO_2_ tank. In order to minimize suffering and pain of the mice, the related experimental procedures were approved by the Animal Ethics Committee of Dalian Polytechnic University (DPU) and conducted in accordance with the Guidelines for Use and Care of Laboratory Animals of DPU.

### 2.3. Oral Administration of A-GAs and Sample Preparation

#### 2.3.1. Preparation of Samples from Gastrointestinal Tract Contents

Mice were divided into three groups: Group 1—G-C4:0 group, mice were gavaged with G-C4:0 (0.30 mmol/kg bodyweight) dissolved in 25% (*w*/*v*, g/mL) polyethylene glycol-6000 (PEG-6000) solution; Group 2—G-C14:0 group, mice were gavaged with G-C14:0 solution (0.30 mmol/kg bodyweight) dissolved in 25% (*w*/*v*, g/mL) PEG-6000 solution; Group 3—control group, mice were gavaged with 25% (*w*/*v*, g/mL) PEG-6000 solution. The animals were then sacrificed by CO_2_ asphyxiation at different time points (0, 1, 2, 4 and 6 h; n = 8 mice/time point), and the contents from the colon, cecum, small intestine and stomach were collected. Finally, the supernatant liquid was obtained after centrifugation (19,000× *g*, 4 °C and 20 min) with a centrifugal machine (Himac CF16RN, Hitachi, Japan). Before HPLC-UV analysis, the samples were filtered through 0.22 μm spin filters.

#### 2.3.2. Preparation of Samples from Fecal Bacteria Fermentation

Freshly-dropped feces collected randomly from KM mice were weighed, and then suspended in 5-fold volume (*v*/*w*, mL/g) of saline solution. After centrifugation (1000× *g*, 4 °C, 5 min), the fecal slurry was separated. The fermentation sample consisted of four components as follows: 264 μL of bacterial culture medium, 3 μL of A-GA (100 mM), 30 μL of mice fecal slurry and 3 μL of tween-20. The mass or volume of components in one liter of bacterial culture medium was as follows: 0.5 g of bile salts, 0.5 g of L-cysteine, 0.01 g of calcium chloride, 2 g of yeast extract, 2 mL of tween-20, 2 g of sodium hydrogen carbonate, 0.04 g of dipotassium phosphate, 2 g of peptone, 10 μL of vitamin K, 2 μL of 1% (*w*/*v*, g/mL) resazurin solution, 0.1 g of sodium chloride, 0.01 g of magnesium sulfate heptahydrate and 0.045 g of monopotassium phosphate [16]. After incubation at 37 °C and nitrogen protection for different times (0, 24, 48 and 72 h), 150 μL of methanol was added to terminate the fermentation reaction. Finally, the supernatant was obtained after centrifugation (19,000× *g*, 4 °C and 20 min). Before HPLC-UV analysis, the samples were filtered through 0.22 μm spin filters.

#### 2.3.3. Preparation of Samples from *Lactobacillus* Fermentation

*Lactobacillus gasseri*, *Lactobacillus johnsonii* and *Lactobacillus reuteri* were involved in our studies, which were isolated from the KM mice [12]. The microbial hydrolysis of A-GAs by *Lactobacillus* was evaluated in an in vitro anaerobic fermentation model as reported in our previous study [13]. In brief, the fermentation sample consisted of four components as follows: 264 μL of *de Man*, *Rogosa*, and *Sharpe* (MRS) medium, 3 μL of A-GA (100 mM), 30 μL of suspensions of *Lactobacillus* (inoculation amount 2%, *v*/*v*, mL/mL) and 3 μL of tween-20. After incubation at 37 °C and nitrogen protection for different times (0, 24, 48 and 72 h), 150 μL of methanol was added to terminate the fermentation reaction. Finally, the supernatant was obtained after centrifugation (19,000× *g*, 4 °C and 20 min). Before HPLC-UV analysis, the samples were filtered through 0.22 μm spin filters.

### 2.4. HPLC-UV Analysis

#### 2.4.1. HPLC-UV Analytical Method

Column: Hypersil GOLD^TM^ column (5 μm, 250 mm × 4.6 mm) (Thermo Scientific, California, USA); equipment: LC-20AVP system (SHIMADZU, Kyoto, Japan); wavelength: 273 nm; column oven: 40 °C; mobile phase A: 0.1% (*v*/*v*, mL/mL) acetic acid in ultrapure water; mobile phase B: methanol; injection volume: 4 μL; elution velocity: 0.8 mL min^−1^; elution program: 80% A (0–9 min), 80-5% A (9–10 min), 5% A (10–30 min), 5-1% A (30–31 min), 1% A (31–35 min), 1–80% A (35–36 min) and 80% A (36–45 min).

#### 2.4.2. The External Standard Curves

The external standard curves of G-C4:0 and G-C14:0 for the quantification of GA and A-GAs in gastrointestinal tract contents were as follows: y = 3,000,000.0x + 75,327.6 (R^2^ = 0.9997) and y = 4,000,000.0x + 54,843.1 (R^2^ = 0.9999), respectively. Meanwhile, the external standard curves of GA for the quantification of GA in fermentation samples of fecal bacteria and *Lactobacillus* were as follows: y = 2001.5x + 8747.7 (R^2^ = 0.9992) and y = 4,000,000.0x + 56,446.8 (R^2^ = 0.9994), respectively. Moreover, the coefficients of variation (precision) were less than 15% for intra-day and inter-day trials, and the accuracy was between 90% and 110% (of true values).

#### 2.4.3. The Degree of A-GAs Hydrolysis in Fermentation Samples

The calculation formula of the degree of A-GAs hydrolysis (DH, %) was as follows:DH = (C_GA_V × 100)/C_0_V_0_
where C_GA_: the concentration of GA in the sample after fermentation, which is calculated using the above-mentioned external standard curve; C_0_: the concentration (100 mM) of A-GAs in the sample before fermentation; V: the volume (450 μL) of the sample after fermentation; V_0_: the volume (3 μL) of added A-GAs before fermentation.

### 2.5. Statistical Analysis

The results were calculated from parallel measurements, which were shown as the means ± standard derivations. The SPSS version 16.0 software (SPSS Inc., Chicago, IL, USA) was applied for the multiple comparisons of means. Briefly, significant differences among several groups were analyzed according to ANOVA and Student–Newman–Keuls (*p* < 0.05).

## 3. Results and Discussion

### 3.1. Concentrations of A-GAs in the Stomach

In the gastric lumen, the concentrations of A-GAs gradually decreased over time (Figure 2A,B). Furthermore, the concentrations of A-GAs in the stomach contents were calculated (Figure 2C). Obviously, the concentration of butyl gallate (G-C4:0) in the gastric contents decreased more slowly. For example, the concentrations of G-C4:0 and myristyl gallate (G-C14:0) in the stomach contents were 0.29 and 0.02 mM at 2 h after oral administration, respectively. It is well known that the surface mucous cells of the stomach make up the gastric pits, which lead to branched, long, tubular glands. These glands give the gastric mucosa a leafy appearance, also known as gastric foveolae [17]. In contrast with G-C14:0 (Log P = 5.66), G-C4:0 (Log P = 1.76) with suitable lipid solubility appear to remain in the gastric foveolae for a longer period of time. Therefore, G-C4:0 is likely to move slowly down the small intestine.

On the other hand, during the retention of A-GAs in stomach, no free GA was detected in all samples. It clearly indicated that A-GAs were stable in the acid environment of the stomach. Our previous studies have demonstrated that tyrosol fatty acid esters (TYr-Es) remained remarkably stable in the acidic environment of the stomach, which is consistent with the experimental results in this section [12,13]. Other lipids such as sterol esters and phospholipids also remain unhydrolyzed in the stomach. Additionally, it is widely known that there is also a certain amount of gastric lipase in the stomach, which can only hydrolyze the triacylglycerols. By contrast, the gastric lipase is not able to hydrolyze lipids including sterol esters and phospholipids [18].

### 3.2. Concentrations of A-GAs in the Small Intestine

In the small intestine lumen, the concentrations of A-GAs gradually decreased over time (Figure 2D,E). Furthermore, the concentrations A-GAs in the small intestine contents were calculated (Figure 2F). Obviously, G-C4:0 decreased more sharply in the small intestine. For example, the G-C4:0 was almost undetectable at 2 h after oral administration, while the corresponding mass concentration of G-C14:0 was 0.56 mM.

The rapid decrease was mainly caused by the transport characteristic of G-C4:0 across the intestinal mucosa. The small intestine is the major site of the absorption of most nutrients from the diet. Take fats (mainly triglycerides) as an example, monoglyceride and free fatty acid digested from triglycerides are absorbed into blood across the mucosa of the small intestine [19]. In our previous study, the short-chain A-GAs including G-C4:0, propyl gallate (G-C3:0), ethyl gallate (G-C2:0) and methyl gallate (G-C1:0) can cross the membrane of the everted rat gut sac model (ERGSM) (i.e., from the mucosal side to the serosal side) in the form of esters. However, no measurable chromatographic peaks corresponding to long-chain A-GAs, including G-C14:0, octyl gallate (G-C8:0), dodecyl gallate (G-C12:0) and hexadecyl gallate (G-C16:0) were observed in the serosal fluids of ERGSM [20].

It showed that the transportation across the mucosa of the small intestine to the blood, as well as the slow movement from the small intestine to the large intestine, lead to the decrease of A-GAs. In addition, the hydrolysis of A-GAs in the small intestine may also contribute to this decrement.

### 3.3. Hydrolysis of A-GAs in the Small Intestine

In the small intestine lumen, free GAs were liberated from G-C4:0 and G-C14:0 during their retention in the small intestine (Figure 2D,E and Figure 3), which clearly indicated that A-GAs can be hydrolyzed by intestinal lipases. In our previous studies, the results in the simulated gastrointestinal digestion model and ERGSM demonstrated that intestinal lipases can hydrolyze phenolipids including A-GAs and TYr-Es to produce GA and TYr, respectively [20,21,22]. In particular, the degree of hydrolysis (DH) of G-C14:0 was significantly lower than that of G-C4:0 (K_C14:0_ < K_C4:0_, Figure 3B), which is consistent with the experimental results in this section. Intestinal lipases are a class of hydrolase, which mainly catalyze the hydrolytic degradation of triglycerides or other esters to produce free fatty acids over an oil–water interface [23]. For A-GAs, compared with a short hydrophilic chain such as C4:0, a long hydrophobic chain such as C14:0 in A-GAs could not ensure a proper interfacial location, which further lead to lower DH [24]. Various studies have also indicated that the rates of hydrolysis of triglycerides (with different fatty acids) by intestinal lipases in increasing order were as follows: C18:0, C16:0, C14:0, C12:0, C10:0 and C8:0 (C4:0) [25]. In addition, compared with the DH values of TYr-Es [22], those of A-GAs were significantly lower. This was due to the fact that, compared with the fatty acid chains in TYr-Es (Figure 1), fatty alcohol chains in A-GAs are harder to bind to the hydrophobic canyon of intestinal lipase [15].

Currently, many studies have reported that GA has various health benefits including enhancing memory ability [26], inhibition of inflammation [27] and antioxidant activity [28]. It should be noted that these health benefits are related most closely to the duration of action [29]. Unfortunately, GA has a short half-life time (in vivo residence time) [30]. Here, it is obvious that A-GAs have a certain slow-release effect, which could increase the GA loading cycle time by prolonging the terminal half-life. Given this, due to the in vivo circulation time, the synthesis of phenolipids such as A-GAs could be a useful approach to enhance the health benefits of polyphenols such as GA.

### 3.4. Concentrations of A-GAs in the Cecum

The gastrointestinal tract consists of the stomach, small intestine (including duodenum, ileum and jejunum) and large intestine (including colon and cecum). Obviously, the large intestine is the last checkpoint before the undigested foods are excreted in the feces. Thus, A-GAs were undetectable in cecum contents before 1 h (Figure 2G,H).

Furthermore, the mass concentrations of A-GAs in the cecum were calculated (Figure 2I). It is worth noting that the concentration of G-C4:0 in the cecum content was lower than that of G-C14:0. For example, the concentrations of G-C4:0 and G-C14:0 in the cecum contents were 0.90 and 3.24 mM at 2 h after oral administration. In our previous study, G-C4:0 has been shown to cross the small intestine membrane in the form of esters in the ERGSM. Meanwhile, compared with G-C14:0, G-C4:0 hydrolyzed more strongly [20]. Therefore, the concentration of G-C4:0 in the cecum is lower, which is mainly due to its stronger hydrolysis and transportation ability in the small intestine. Many in vivo studies have demonstrated that A-GAs such as dodecyl gallate (G-C12:0), propyl gallate (G-C3:0) and methyl gallate (G-C1:0) exert stronger beneficial effects in the pathological models of intestinal disorders when compared to those of GA [7]. Given this, the ability to reach the cecum will lead to a strong positive impact of A-GAs on intestinal health.

### 3.5. Concentrations of A-GAs in the Colon

A-GAs were not detected in colon contents at 0 h (Figure 2J,K). Surprisingly, 1 h after oral administration, a detectable level of A-GAs existed in the colon. The mass concentrations of A-GAs in the colon were calculated (Figure 2L).

In the gastrointestinal tract, A-GAs undergo the metabolic reaction by gut microbiota. The microbiota in duodenum and jejunum is mainly consisted of various acid-tolerant bacteria such as *Lactobacilli* and *Streptococci* [31]. By contrast, the cecum and colon mainly contain anaerobic and aerobic bacteria [32]. Some bioactive compounds which are metabolized by microbiota in the cecum and colon will have stronger beneficial effects on health [33]. Taking oleuropein and total secoiridoid (two glycosidic ligand structures of hydroxytyrosol) as examples, they undergo microbial metabolic reactions in the gastrointestinal tract. The related metabolites have numerous beneficial effects on health [34]. Notably, unabsorbed A-GAs can transport GA in the form of an ester to the cecum and colon, which will undergo metabolic degradation and further contribute to various health-promoting effects.

### 3.6. Microbial Hydrolysis Characteristic of A-GAs by Mice Fecal Microbiota

It was obvious that high levels of A-GAs were able to reach the colon and cecum. On the one hand, A-GA itself can repair damaged intestine. On the other hand, microbial esterase from gut microbiota in the colon and cecum can hydrolyze the ester bond to liberate GA. Therefore, the microbial hydrolysis characteristic of A-GAs was evaluated by using an in vitro fermentation model. Under the action of fecal microbiota, free GA was liberated from A-GAs (Figure 4). The time-dependent manner clearly indicated the occurrence of hydrolysis. Generally, a larger K_DH_ indicates more a rapid hydrolysis. Therefore, consistent with our previous findings obtained by lipase in ERGSM [16], with the increase in the alkyl chain length, the DHs of A-GAs initially increased, peaking at G-C8:0 and then decreased. For example, 48 h after fermentation, the DHs of G-C1:0, G-C2:0, G-C3:0, G-C4:0, G-C8:0, G-C12:0, G-C14:0, G-C16:0 and G-C18:0 were 1.44, 1.76, 1.54, 2.13, 3.12, 2.83, 2.58, 2.24 and 1.71%, respectively. Generally, fecal microbiota lipase carries out a heterogeneous catalysis, and the catalytic activity is maximal only when the interaction of lipase is at an oil–water interface [35]. Usually, long-chain esters with high hydrophobicity and short-chain esters with high hydrophilicity do not easily form an oil–water emulsion [36]. By contrast, medium-chain esters containing C8:0 are prone to form an emulsion, which leads to a higher degree of hydrolysis.

The gut microbiota in the colon and cecum play important roles in the digestion of nutrients and the body health [37]. They are able to secrete extracellular lipases including fatty-acid-specific, 1,3-specific and non-specific lipases. Most of these lipases secreted from gut microbiota belong to nonspecific lipases. For example, the lipase secreted from *Lactobacillus helveticusx* is able to hydrolyze methyl p-coumarate and methyl ferulate [38]. The lipase secreted from *Staphylococcus aureus* mainly hydrolyzes acyl esters of p-nitrophenol (or umbelliferone) and short-chain triacylglycerols [39]. Furthermore, the extracellular tannase secreted from *Lactobacillus plantarum* and *Lactobacillus subtilis AM1* can catalyze the cleavage of ester bonds in various compounds [40]. Therefore, A-GAs are able to deliver GA and fatty alcohols to the colon and cecum, which can further participate in the regulation of gut microbiota [41].

### 3.7. Microbial Hydrolysis Characteristics of A-GAs by Lactobacillus

Among the various species of gut microbiotas, *Lactobacillus* is an important bacterial specie in the gastrointestinal tract. So far, strains of *Lactobacillus* have been reported to produce lipase to hydrolyze the ester bonds. In order to further examine the effects of *Lactobacillus* in the microbial hydrolysis of A-GAs, *Lactobacillus gasseri*, *Lactobacillus reuteri* and *Lactobacillus johnsonii* were isolated from the feces of mice. The DHs of A-GAs with various alkyl chain lengths were evaluated in the in vitro fermentation model by HPLC-UV. As shown in Figure 5 and Figure 6, all of the A-GAs were hydrolyzed by the above three *Lactobacillus* to free GA. Similar to the hydrolysis regularity by the fecal microbiota in the previous chapter, the DHs of A-GAs initially increased, peaking at G-C8:0 and then decreased with the increase in the alkyl chain length. For example, after incubating at 37 °C for 72 h, the DHs of G-C1:0, G-C2:0, G-C3:0, G-C4:0, G-C8:0, G-C12:0, G-C14:0, G-C16:0 and G-C18:0 by *L. reuteri* were 0.80, 0.75, 0.70, 0.89, 1.15, 0.98, 0.88, 0.84 and 0.84%, respectively.

In this study, A-GAs were hydrolyzed by *L. gasseri*, *L. reuteri* and *L. johnsonii* to produce free GA. Many studies have demonstrated that, as a kind of microbial-derived metabolite from dietary polyphenols, GA can participate in the regulation of the composition of the gut microbiota, and further exert beneficial effects in gut health [42]. Meanwhile, after microbial hydrolysis by *Lactobacillus* in the gastrointestinal tract, A-GAs can also deliver fatty alcohols to the large intestine. These fatty alcohols exhibit a wide range of biological activities including the regulation of intestinal microbiota and neuroprotective effects [41].

## 4. Conclusions

Oral alkyl gallates (A-GAs) remain stable in the stomach environment of mice. The lipases in the small intestine are able to hydrolyze A-GAs, which shows the desired release behavior of GA. Most importantly, A-GAs can reach the cecum and the colon, which exhibits a beneficial impact on intestinal health. Furthermore, A-GAs could also be hydrolyzed by the gut microbiota including *Lactobacillus gasseri*, *Lactobacillus reuteri* and *Lactobacillus johnsonii* to produce free GA. Overall, this study suggested that the formation of A-GAs will provide a useful new approach to sustained-release of GA. Of course, more studies are needed to clarify the potential enhanced intestinal health benefits, which will effectively expand the application of A-GAs in the field of functional food and biomedicine.

## Figures and Tables

**Figure 1 foods-11-03936-f001:**
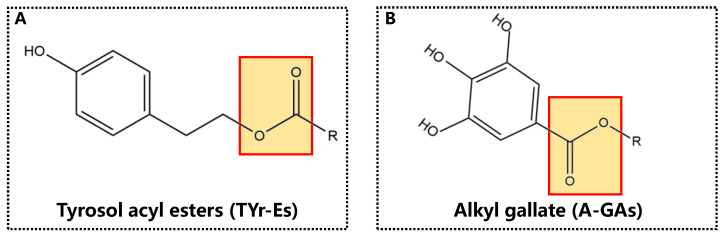
Structural formulas of tyrosol acyl esters (**A**) and alkyl gallates (**B**).

**Figure 2 foods-11-03936-f002:**
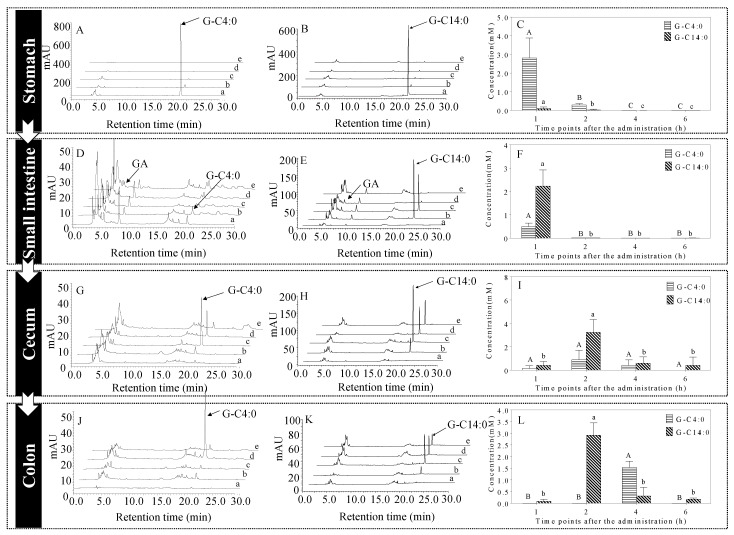
HPLC-UV chromatograms (G-C4:0 group-(**A**,**D**,**G**,**J**); G-C14:0 group-(**B**,**E**,**H**,**K**)) and the concentrations of G-C4:0 and G-C14:0 in the contents of the stomach (**C**), small intestine (**F**), cecum (**I**) and colon (**L**) (n = 8): 0, 1, 2, 4 and 6 h (a–e, respectively). Bars with different letters (A–C, a–c) are statistically different (*p* < 0.05).

**Figure 3 foods-11-03936-f003:**
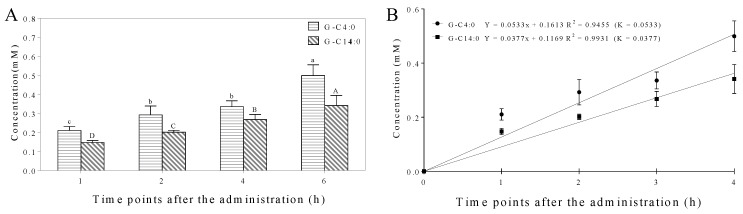
Concentrations (**A**) and kinetic models (**B**) of GA liberated from the hydrolysis of G-C4:0 and G-C14:0 in the contents of small intestine. Bars with different letters (A–D, a–c) are statistically different (*p* < 0.05).

**Figure 4 foods-11-03936-f004:**
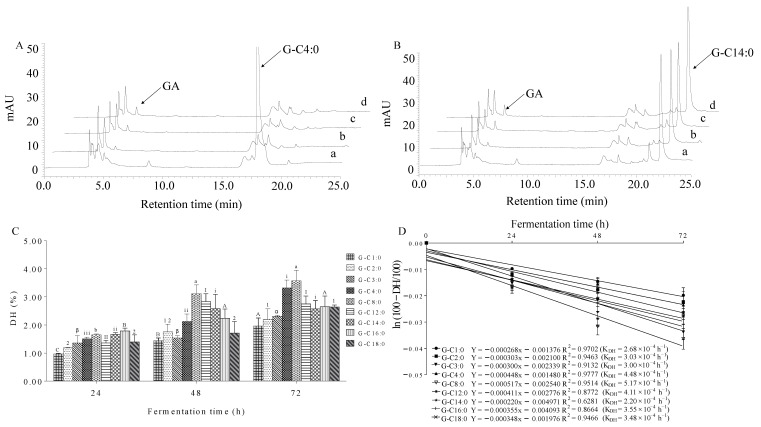
HPLC-UV chromatograms (G-C4:0 group-(**A**) and G-C14:0 group-(**B**)), DHs (**C**) and kinetic models (**D**) of GA liberated from the hydrolysis of A-GAs by mice fecal microbiota: 0, 24, 48 and 72 h (a–d, respectively). Bars with different symbols (A–C, a–b, α–β, i–iii, I–II, 1–2) are statistically different (*p* < 0.05).

**Figure 5 foods-11-03936-f005:**
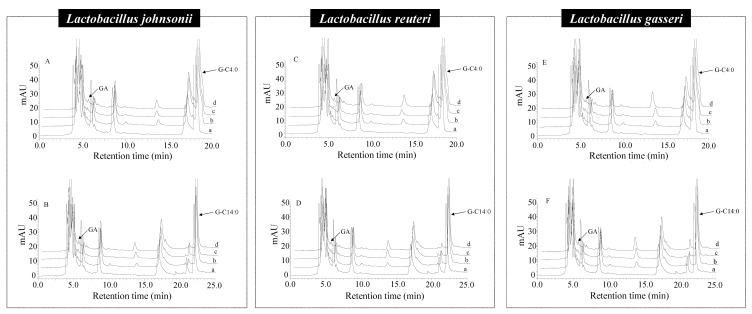
HPLC-UV chromatograms of GA liberated from G-C4:0 (**A**,**C**,**E**) and G-C14:0 (**B**,**D**,**F**) by *Lactobacillus johnsonii*, *Lactobacillus reuteri* and *Lactobacillus gasseri*: 0, 24, 48 and 72 h (a–d, respectively).

**Figure 6 foods-11-03936-f006:**
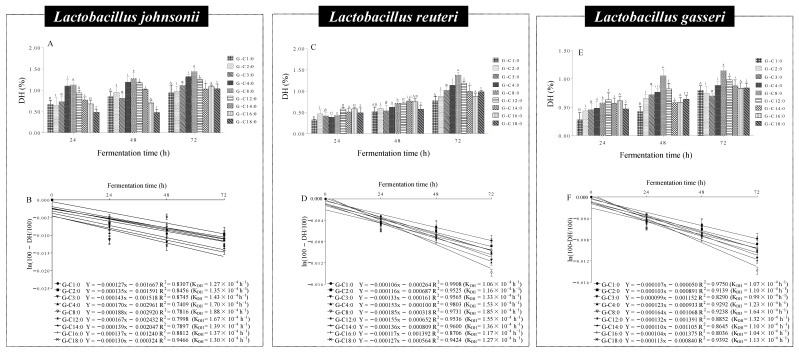
DHs (**A**,**C**,**E**) and kinetic models (**B**,**D**,**F**) of GA liberated from G-C4:0 and G-C14:0 by *Lactobacillus johnsonii*, *Lactobacillus reuteri* and *Lactobacillus gasseri*. Bars with different letters (A–B, a–c, α–β, i–ii, I–II, 1–2) are statistically different (*p* < 0.05).

## Data Availability

The data presented in this study are available on request from the corresponding author.

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
