# Peer review of "Gastrointestinal Digestion and Microbial Hydrolysis of Alkyl Gallates: Potential Sustained Release of Gallic Acid"

_foods, 2022, doi:10.3390/foods11233936_

Round 1

Reviewer 1 Report

please see the attached file

Author Response

Dear reviewer:

Re: foods-2013764

We are grateful to you for constructive comments and some important changes requested in our manuscript. We have accordingly carefully revised our manuscript to address all the comments made. The main revisions are explained below as “Response to Reviewer 1’s Comments”. We trust that our manuscript is now ready for further processing for publication in the “Foods”.

Reviewer #1:

Comments:

  1. In the title, remove “the enhance intestinal protective effect” because there was no result can support this effect.

Response: Thank you very much for your suggestion. According to your guideline, the title has been changed to "Gastrointestinal digestion and microbial hydrolysis of alkyl gallates: Potential sustained release of gallic acid ". Please read Page 1, Lines 2-3.

  1. “In vivo”, “in vitro” should be italic

Response: Thank you very much for your carefully review. In the revised manuscript, all related details have been corrected marked with font colors. The relevant screenshots are shown as follows. Please read Page 1, Line 40; Page 3, Line 132; Page 7, Lines 242-244; Page 7, Line 259; Page 7, Line 284; Page 8, Line 314; Page 13, Line 410.

  1. Line135, what is the concentration for the lactobacillus?

Response: Thank you very much for your carefully review. Lactobacillus johnsonii, Lactobacillus reuteri and Lactobacillus gasseri were used in this study. These strains were isolated in our laboratory from the mice. Briefly, the mice feces were properly diluted, and further plated on de Man & Rogosa & Sharpe (MRS) agar plates containing 3% (w/v, g/mL) calcium carbonate (CaCO3). The strains were grown anaerobically on plates for 48 h at 37 °C, and the isolates yielded a dissolving circle of CaCO3. Gram staining and microscopic observation were used to identify the potential strains of Lactobacilli. The isolated Lactobacilli were grown in MRS medium for 12 h at 37 °C under anaerobic conditions. Then, under the same conditions (inoculation amount 2%, v/v, mL mL-1), these isolated Lactobacilli were transferred three times. According to your guideline, the concentrations of Lactobacilli have been added in the revised manuscript. To be specific, "In brief, the fermentation sample consisted of four components as follows: 264 μL of de Man, Rogosa, and Sharpe (MRS) medium, 3 μL of A-GA (100 mM), 30 μL of suspensions of Lactobacillus and 3 μL of tween-20." has been changed to "In brief, the fermentation sample consisted of four components as follows: 264 μL of de Man, Rogosa, and Sharpe (MRS) medium, 3 μL of A-GA (100 mM), 30 μL of suspensions of Lactobacillus (inoculation amount 2%, v/v, mL/mL) and 3 μL of tween-20." marked with font colors. Please read Page 3, Lines 133-136.

  1. Figure legend for figure 5 is wrong. Current one is figure legend for figure 4.

Response: We were really sorry for our careless mistakes. Accordingly, the legend of Figure 5 has been changed to "Figure 5. HPLC-UV chromatograms of GA liberated from G-C4:0 (A, C and E) and G-C14:0 (B, D and F) by Lactobacillus johnsonii, Lactobacillus reuteri and Lactobacillus gasseri: 0, 24, 48 and 72 h (a-d, respectively)." in the revised manuscript. Please read Page 10, Lines 346-347.

  1. Figure 5 C, D, G, H, K, L is too small to distinguish.

Response: Thank you very much for your carefully review. According to your guideline, the quality of Figure 5 has been improved in the revised manuscript. Besides, Figure 5 has been divided to Figure 5 and Figure 6. The new figures are shown as follows. Please read Page 10, Lines 345-351.

Figure 5. HPLC-UV chromatograms of GA liberated from G-C4:0 (A, C and E) and G-C14:0 (B, D and F) by Lactobacillus johnsonii, Lactobacillus reuteri and Lactobacillus gasseri: 0, 24, 48 and 72 h (a-d, respectively).

Figure 6. DHs (A, C and E) and kinetic models (B, D and F) of GA liberated from G-C4:0 and G-C14:0 by Lactobacillus johnsonii, Lactobacillus reuteri and Lactobacillus gasseri.

Different letters indicate significant differences (P < 0.05)

Reviewer 2 Report

The information described in the manuscript is interesting, however, it is necessary to address the following comments:

Line 96: remove alkyl gallates…GA or A-Gas. At…

Line 110: n=8 or n = 8, like in line 195?

Line 111: indicate the equipment model, trademark, and country (centrifuge)

Line 112: indicate the temperature conditions

Line 112: indicate the equipment model, trademark, and country (HPLC system)

Line 112: After centrifugation (19,000 x g/??? °C/20 min)

Line 116: After centrifugation (1,000 x g/4 °C/5 min)

Line 127: After centrifugation (19,000 x g/??? °C/20 min)

Line 132: change to italic text format (in vitro)

Line 138: After centrifugation (19,000 x g/??? °C/20 min)

Line 150: remove space (x +75327.6)

Line 172: remove alkyl gallates

Line 196: modify text size format

Line 224: change [20-22] by [20–22]

Line 240,242,263: change to italic text format (in vivo)

Line 290,326,418: change to italic text format (in vitro)

Line 308: modify text size format

Line 332: modify by L. reuteri

Line 334: modify by L. gasseri, L. reuteri and L. johnsonii

Line 343: the size of the figure should be increased, due to the fact that the information included is not appreciated

Line 347: modify text size format

Line 376: according to the author guide the article volume should be appear in italic text format, modify in the required references

Line 377: modify… why?. Bio

Line 377: modify…Biochimie 2012

Line 396: Multifunctional Gut of Fish (journal abbreviation?)

Line 409: modify by…. Nutrition 2012

Line 413: Curr. Drug Deliv.

Line 429: Synapse 2020

Line 437: Journal of Shenyang Pharmaceutical University (journal abbreviation?)

Line 443: Chromatographia 2016

Line 457: Nature 2010

Author Response

Dear reviewer:

Re: foods-2013764

We are grateful to you for constructive comments and some important changes requested in our manuscript. We have accordingly carefully revised our manuscript to address all the comments made. The main revisions are explained below as “Response to Reviewer 2’s Comments”. We trust that our manuscript is now ready for further processing for publication in the “Foods”.

Reviewer #2:

Comments:

  1. Line 96: remove alkyl gallates…GA or A-Gas. At…

Response: Thank you very much for your carefully review. According to your guideline, the related details have been corrected. The relevant screenshots are shown as follows. Please read Page 3, Line 96.

  1. Line 110: n=8 or n = 8, like in line 195?

Response: Thank you very much for your carefully review. According to your guideline, the related details have been corrected. The relevant screenshots are shown as follows. Please read Page 3, Line 110.

  1. Line 111: indicate the equipment model, trademark, and country (centrifuge)

Response: Thank you very much for your carefully review. According to your guideline, the related details have been corrected. The relevant screenshots are shown as follows. Please read Page 3, Line 112.

  1. Line 112: indicate the temperature conditions

Response: Thank you very much for your carefully review. According to your guideline, the related details have been corrected. The relevant screenshots are shown as follows. Please read Page 3, Line 112.

  1. Line 112: indicate the equipment model, trademark, and country (HPLC system)

Response: Thank you very much for your carefully review. According to your guideline, the related details have been corrected. The relevant screenshots are shown as follows. Please read Page 3, Lines 140-144.

  1. Line 112: After centrifugation (19,000 x g/??? °C/20 min)

Response: Thank you very much for your carefully review. According to your guideline, the related details have been corrected. The relevant screenshots are shown as follows. Please read Page 3, Line 112.

  1. Line 116: After centrifugation (1,000 x g/4 °C/5 min)

Response: Thank you very much for your carefully review. According to your guideline, the related details have been corrected. The relevant screenshots are shown as follows. Please read Page 3, Lines 116-117.

  1. Line 127: After centrifugation (19,000 x g/??? °C/20 min)

Response: Thank you very much for your carefully review. According to your guideline, the related details have been corrected. The relevant screenshots are shown as follows. Please read Page 3, Lines 126-128.

  1. Line 132: change to italic text format (in vitro)

Response: Thank you very much for your carefully review. According to your guideline, the related details have been corrected. The relevant screenshots are shown as follows. Please read Page 3, Line 132.

  1. Line 138: After centrifugation (19,000 x g/??? °C/20 min)

Response: Thank you very much for your carefully review. According to your guideline, the related details have been corrected. The relevant screenshots are shown as follows. Please read Page 3, Lines 137-138.

  1. Line 150: remove space (x +75327.6)

Response: Thank you very much for your carefully review. According to your guideline, the related details have been corrected. The relevant screenshots are shown as follows. Please read Page 4, Lines 149-151.

  1. Line 172: remove alkyl gallates

Response: Thank you very much for your carefully review. According to your guideline, the related details have been corrected. The relevant screenshots are shown as follows. Please read Page 4, Line 172.

  1. Line 196: modify text size format

Response: Thank you very much for your carefully review. According to your guideline, the related details have been corrected. The relevant screenshots are shown as follows. Please read Page 5, Line 195.

  1. Line 224: change [20-22] by [20–22]

Response: Thank you very much for your carefully review. According to your guideline, the related details have been corrected. The relevant screenshots are shown as follows. Please read Page 6, Lines 222.

  1. Line 240,242,263: change to italic text format (in vivo)

Response: Thank you very much for your carefully review. According to your guideline, the related details have been corrected. The relevant screenshots are shown as follows. Please read Page 6, Lines 242-244; Please read Page 7, Lines 259-262.

  1. Line 290,326,418: change to italic text format (in vitro)

Response: Thank you very much for your carefully review. According to your guideline, the related details have been corrected. The relevant screenshots are shown as follows. Please read Page 7, Lines 283-284; Page 8, Lines 313-316; Page 13, Lines 418-419.

  1. Line 308: modify text size format

Response: Thank you very much for your carefully review. According to your guideline, the related details have been corrected. The relevant screenshots are shown as follows. Please read Page 9, Lines 342-344; Page 11, Lines 349-351.

  1. Line 332: modify by Lreuteri

Response: Thank you very much for your carefully review. According to your guideline, the related details have been corrected. The relevant screenshots are shown as follows. Please read Page 8, Lines 318-329.

  1. Line 334: modify by gasseri, L. reuteri and L. johnsonii

Response: Thank you very much for your carefully review. According to your guideline, the related details have been corrected. The relevant screenshots are shown as follows. Please read Page 8, Lines 318-329.

  1. Line 343: the size of the figure should be increased, due to the fact that the information included is not appreciated

Response: Thank you very much for your carefully review. According to your guideline, the new figures has been redrawn. The new figures are shown as follows. Please read Page 10, Lines 345-351.

Figure 5. HPLC-UV chromatograms of GA liberated from G-C4:0 (A, C and E) and G-C14:0 (B, D and F) by Lactobacillus johnsonii, Lactobacillus reuteri and Lactobacillus gasseri: 0, 24, 48 and 72 h (a-d, respectively).

Figure 6. DHs (A, C and E) and kinetic models (B, D and F) of GA liberated from G-C4:0 and G-C14:0 by Lactobacillus johnsonii, Lactobacillus reuteri and Lactobacillus gasseri.

Different letters indicate significant differences (P < 0.05)

  1. Line 347: modify text size format

Response: Thank you very much for your carefully review. According to your guideline, the related details have been corrected. The relevant screenshots are shown as follows. Please read Page 11, Lines 349-351.

  1. Line 376: according to the author guide the article volume should be appear in italic text format, modify in the required references

Response: Thank you very much for your carefully review. According to your guideline, the related details have been corrected. The relevant screenshots are shown as follows. Please read Page 12, Lines 370-457.

  1. Line 377: modify… why?. Bio

Response: Thank you very much for your carefully review. According to your guideline, the related details have been corrected. The relevant screenshots are shown as follows. Please read Page 12, Lines 370-371.

  1. Line 377: modify…Biochimie2012

Response: Thank you very much for your carefully review. According to your guideline, the related details have been corrected. The relevant screenshots are shown as follows. Please read Page 12, Lines 370-371.

  1. Line 396: Multifunctional Gut of Fish (journal abbreviation?)

Response: Thank you very much for your carefully review. According to your guideline, the related details have been corrected. The relevant screenshots are shown as follows. Please read Page 12, Lines 389-390.

  1. Line 409: modify by…. Nutrition2012

Response: Thank you very much for your carefully review. According to your guideline, the related details have been corrected. The relevant screenshots are shown as follows. Please read Page 12, Lines 401-402.

  1. Line 413:  Drug Deliv.

Response: Thank you very much for your carefully review. According to your guideline, the related details have been corrected. The relevant screenshots are shown as follows. Please read Page 13, Line 406.

  1. Line 429: Synapse2020

Response: Thank you very much for your carefully review. According to your guideline, the related details have been corrected. The relevant screenshots are shown as follows. Please read Page 13, Lines 420-422.

  1. Line 437: Journal of Shenyang Pharmaceutical University(journal abbreviation?)

Response: Thank you very much for your carefully review. According to your guideline, the related details have been corrected. The relevant screenshots are shown as follows. Please read Page 13, Lines 430-431.

  1. Line 443: Chromatographia2016

Response: Thank you very much for your carefully review. According to your guideline, the related details have been corrected. The relevant screenshots are shown as follows. Please read Page 12, Lines 435-436.

  1. Line 457: Nature2010

Response: Thank you very much for your carefully review. According to your guideline, the related details have been corrected. The relevant screenshots are shown as follows. Please read Page 13, Lines 448-450.

Reviewer 3 Report

The manuscript titled to Gastrointestinal digestion and microbial hydrolysis of alkyl gallates: Potential sustained release of gallic acid and enhance intestinal protective effect was critically reviewed. The purpose of the study should be written more clearly and clearly both in the abstract and in the introduction. More information can be given about the main problems that led to conduct to this study. The manuscript was well written except discussion section and conclusion. The authors should discuss their results with more up-to-date sources and in a more critical and rigorous manner. The conclusions of the study should be strengthened, the limitations and strengths of the study should be detailed, and recommendations should be developed for the adaptation of the results of this study in routine clinical practice.

Author Response

Dear reviewer:

Re: foods-2013764

We are grateful to you for constructive comments and some important changes requested in our manuscript. We have accordingly carefully revised our manuscript to address all the comments made. The main revisions are explained below as “Response to Reviewer 3’s Comments”. We trust that our manuscript is now ready for further processing for publication in the “Foods”.

Reviewer #3:

Comments:

  1. The manuscript titled to Gastrointestinal digestion and microbial hydrolysis of alkyl gallates: Potential sustained release of gallic acid and enhance intestinal protective effect was critically reviewed. The purpose of the study should be written more clearly and clearly both in the abstract and in the introduction. More information can be given about the main problems that led to conduct to this study. The manuscript was well written except discussion section and conclusion. The authors should discuss their results with more up-to-date sources and in a more critical and rigorous manner. The conclusions of the study should be strengthened, the limitations and strengths of the study should be detailed, and recommendations should be developed for the adaptation of the results of this study in routine clinical practice.

Response: Thank you very much for your carefully review. According to your guideline, the related details have been improved in the revised manuscript. The relevant screenshots with red are shown as follows. Please read Page 4, Lines 183-191; Page 6, Lines 202-224; Page 7, Lines 264-283; Page 8, Lines 330-340.

Round 2

Reviewer 3 Report

-